# Association between Leukemia Inhibitory Factor Gene Polymorphism and Clinical Outcomes among Young Women with Poor Ovarian Response to Assisted Reproductive Technology

**DOI:** 10.3390/jcm12030796

**Published:** 2023-01-19

**Authors:** Yung-Liang Liu, Chun-I Lee, Chung-Hsien Liu, En-Hui Cheng, Shun-Fa Yang, Hsueh-Yu Tsai, Maw-Sheng Lee, Tsung-Hsien Lee

**Affiliations:** 1Department of Obstetrics and Gynecology, Chung Shan Medical University Hospital, Taichung 40203, Taiwan; 2Department of Obstetrics and Gynecology, Tri-Service General Hospital, National Defense Medical Center, No. 161, Sec. 6, Minquan E. Rd., Neihu Dist., Taipei 11490, Taiwan; 3School of Medicine, Chung Shan Medical University, Taichung 40203, Taiwan; 4Division of Infertility Clinic, Lee Women’s Hospital, Taichung 40602, Taiwan; 5Institute of Medicine, Chung Shan Medical University, Taichung 40203, Taiwan

**Keywords:** polymorphisms, leukemia inhibitory factor, leukemia inhibitory factor, FSH receptor, poor responders, POSEIDON criteria

## Abstract

Background: Does the presence of single-nucleotide polymorphisms (SNPs) in the leukemia inhibitory factor (LIF) gene affect ovarian response in infertile young women? Methods: This was a case–control study recruiting 1744 infertile women between January 2014 to December 2015. The 1084 eligible patients were stratified into four groups using the POSEIDON criteria. The gonadotropin-releasing hormone receptor (GnRHR), follicle-stimulating hormone receptor (FSHR), anti-Müllerian hormone (AMH), and LIF SNP genotypes were compared among the groups. The distributions of LIF and FSHR among younger and older patients were compared. Clinical outcomes were also compared. Results: The four groups of poor responders had different distributions of SNP in LIF. The prevalence of LIF genotypes among young poor ovarian responders differed from those of normal responders. Genetic model analyses in infertile young women revealed that the TG or GG genotype in the LIF resulted in fewer oocytes retrieved and fewer mature oocytes relative to the TT genotypes. In older women, the FSHR SNP genotype contributed to fewer numbers of mature oocytes. Conclusions: LIF and FSHR SNP genotypes were associated with a statistically significant reduction in ovarian response to controlled ovarian hyperstimulation in younger and older women with an adequate ovarian reserve, respectively.

## 1. Introduction

Leukemia inhibitory factor (LIF) is a cytokine belonging to the interleukin-6 superfamily. It was first identified for its ability to induce macrophage differentiation of murine myeloid leukemia cells and inhibit their proliferation [1,2]. LIF modulates various functions and plays important roles in embryo implantation and in non-hormonal contraception [3]. LIF expression has been detected in human follicular fluid and ovarian stromal cells [4]. In animal models, LIF has been reported to enhance the primordial to primary follicle transition of in vitro cultured rat ovaries [5] and promote bovine oocyte maturation for the in vitro maturation of denuded bovine oocytes [6,7], indicating LIF’s potential involvement in folliculogenesis. We hypothesize that LIF is involved in ovarian sensitivity to gonadotropin stimulation, particularly in individuals exhibiting an unexpectedly poor or suboptimal response.

Oocyte quantity and quality are crucial for fecundability [8]. Oocyte quantity can be measured using ovarian reserve markers, including serum anti-Müllerian hormone (AMH) levels and antral follicle count (AFC). The assessment of either serum AMH levels or AFC to determine ovarian response following controlled ovarian hyperstimulation (COH) in women with infertility using artificial reproductive technologies (ARTs) is reported to have sufficient predictive accuracy [9,10]. Ovarian reserve marker-based individualized follicle stimulating hormone (FSH) dosing in women with predicted hyper responders (AFC > 15) undergoing ART can reduce the occurrence of ovarian hyperstimulation syndrome (OHSS). Women using this procedure exhibit typical cumulative live birth rates [11].

Maternal age-related aneuploidy and euploidy affecting oocyte quality are another set of factors that influence ART prognosis [12,13]. However, some patients whose adequate ovarian reserve parameters are at acceptable levels (AFC ≥ 5; AMH ≥ 1.2 ng/mL) exhibit an unexpectedly poor or suboptimal ovarian response after standard ovarian stimulation, resulting in a lower cumulative delivery rate than that of normal responders [14]. These patients were categorized into group 1 (young patients < 35 years) or group 2 (older patients ≥ 35 years) on the basis of the POSEIDON criteria [15]. These unexpectedly poor or suboptimal ovarian responses indicate the limitations of ovarian reserve markers as indicators of ovarian sensitivity to gonadotropin stimulation.

Other candidate biomarkers for ovarian response during ART have been investigated, such as single-nucleotide polymorphisms (SNPs) in the FSH beta-subunit-encoding gene (*FSHB*; [16,17,18], FSH receptor gene (*FSHR*; [18,19,20], and a common polymorphic allele of the LH beta-subunit gene (LH-β variant: *v-βLH*; [21]. In addition to FSH, FSHR and LH are also crucial molecules for ovarian stimulation and function. One study reported the presence of *FSHR* SNPs (rs6165, rs6166, rs1394205) in predicted ovarian normal response, but the clinical relevance of these biomarkers remains minimal [18]. This indicates that other genes related to folliculogenesis or steroidogenesis may be involved in ovarian sensitivity to gonadotropin stimulation.

We performed a case–control study investigating the influence of four SNPs (*LIF*, *AMH*, *FSHR*, and *GnRHR*) on stimulation phase ovarian response and clinical outcomes among patients with infertility undergoing their first ovarian stimulation for in vitro fertilization (IVF) or intracytoplasmic sperm injection (ICSI).

## 2. Materials and Methods

### 2.1. Study Design and Setting

This is a case–control cross sectional study including couples with infertility undergoing their first ovarian stimulation for IVF or ICSI at any period from January 2014 to December 2015. Patients were recruited from Lee Womens’ Hospital in Taichung, Taiwan. The study protocol was approved by the Institutional Review Board of Chung Shan Medical University Hospital (CS13194). The clinical trial registration number was ISRCTN12768989. Written informed consent was obtained from all participants. A venous blood sample was drawn on the day of oocyte retrieval for DNA extraction and subsequent genotyping. The storage DNA samples were genotyped for the present survey. This was also approved by the Institutional Review Board of Chung Shan Medical University Hospital (CS122008).

### 2.2. Patient Selection Criteria

A total of 1744 patients undergoing their first ART cycle were recruited for this study. Among them, 1084 patients fulfilled the POSEIDON criteria of low-prognosis patients in ART. Inclusion criteria were as follows: (1) age ≤ 45 years old (range: 22–45 years), (2) no history of ovarian surgery or pelvic radiation therapy, (3) Han Chinese ethnicity. Exclusion criteria were genetic anomalies, autoimmune dysfunction, inflammatory disease, and other systemic disorders.

### 2.3. Stimulation Protocol

All patients underwent ovarian stimulation followed by oocyte retrieval in a long GnRH agonist stimulation protocol, which has been previously described [22]. All women were treated with fixed daily subcutaneous injections of 0.5 mg of leuprolide acetate (Lupron; Takeda Pharmaceutics, Konstantz, Germany) from day 21 of the previous cycle. A recombinant FSH (Gonal-F, Merck-Serono, Darmstadt, Germany) or highly purified FSH (Menopur; Ferring Pharmaceuticals, Saint-Prex, Switzerland) was administered through an individual set with flexible doses on cycle day 2 or 3. In addition, 10,000 IU human chorionic gonadotropin (Profasi; Serono, Norwell, MA, USA) was administrated to trigger final oocyte maturation and oocyte retrieval was carried out 36 to 38 h thereafter. Fertilization was performed either with conventional insemination or ICSI based on the corresponding semen parameters.

### 2.4. Blood Sampling and DNA Sequencing

Genomic DNA extraction was conducted from ethylenediaminetetraacetic (EDTA) acid anticoagulated venous blood using a QIAamp DNA blood mini kit (Qiagen, Valencia, CA, USA), according to the manufacturer’s instructions [23]. We dissolved DNA in a Tris-EDTA (TE) buffer (10 mM Tris and 1 mM EDTA acid; pH 7.8) and measured the optical density at 260 nm to determine the DNA quantity. The final solution was collected and stored at −20 ℃ until they were used as templates for a polymerase chain reaction (PCR). The genotyping of the four SNPs was performed using the ABI StepOne Real-Time PCR System (Applied Biosystems, Foster City, CA, USA), and allele discrimination was determined using SDS version 3.0 software (Applied Biosystems) and the TaqMan assay (Applied Biosystems) [24]. Table 1 presents the primer sequences we evaluated for each genotype.

### 2.5. Statistical Analysis

We used a chi-squared test to determine the Hardy–Weinberg equilibrium, including *LIF* (rs929271), *AMH* (rs10407022), *FSHR* (rs6166), and *GnRHR* (rs3756159). A chi-square test was used to investigate the associations among POSEIDON groups and tested SNPs under the genotypic (AA versus Aa versus aa) and the allelic (A versus a) models. The recessive (AA versus Aa/aa) model was used for comparison of clinical parameters between various genotype groups. The distribution variables, including demographic characteristics and ovarian response clinical parameters, were determined using a Kolmogorov–Smirnov test. Categorical variables are presented in terms of frequency and percentage, and the continuous variables are presented in terms of median and interquartile range (25th–75th percentile). The Mann–Whitney U test (for continuous variables) or chi-squared test (for categorical items) was applied to evaluate the differences between groups with genetic variants under the recessive model (AA vs. Aa + aa). All data were analyzed using SPSS Statistics for Windows, Version 22.0 (IBM Corp., Armonk, NY, USA). Significance was indicated if *p* < 0.05.

## 3. Results

### 3.1. Patient Baseline Characteristics

In total, 1744 patients who had undergone their first IVF cycles were included, and 1084 patients were stratified on the basis of the POSEIDON criteria into group 1 (*n* = 208), group 2 (*n* = 361), group 3 (*n* = 117), and group 4 (*n* = 398). The other 660 patients, classified as normal responders, constituted the control groups (age ≥ 35 years, *n* = 269; age < 35, *n* = 391).

### 3.2. Genotyping and Polymorphism Analysis

The primers used for each genotype are detailed in Table 1 for *GnRHR* (rs3756159), *FSHR* (rs6166), *AMH* (rs10407022), and *LIF* (rs929271).

The distributions of SNPs for the groups are presented in Table 2. In the comparison of the genotypes across the four POSEIDON groups, significant differences in *LIF* (rs929271) SNP were observed. *LIF* (rs929271) TG plus GG was more common among young women in group 1 (69.2% [53.8% plus 15.4%]) and 3 (64.1% [42.7% plus 21.4%]) than in the older women in group 2 (57.4% [43.8% plus 13.6%]) and 4 (59% [48.2% plus 10.8%]; *p* = 0.0100). No such result was observed for other SNP genotypes, such as *GnRHR* (rs3756159), *FSHR* (rs6166), and *AMH* (rs10407022) (Table 2).

*FSHR* (rs6166) A allele frequencies were higher in women with adequate ovarian reserves and with a suboptimal or poor response after conventional COH (group 1 [69.7%] and 2 [69.4%]) than women with a poor ovarian reserve (group 3 [65.4%] and 4 [63.4%]; *p* = 0.0453). The distribution of G allele frequencies of *LIF* (rs929271) were significantly higher among young women in group 1 (42.3%) and 3 (42.7%) than among older women in group 2 (35.3%) and 4 (34.9%; *p* = 0.0156). No such result was observed for allele frequencies of other SNPs, such as *GnRHR* (rs3756159) and *AMH* (rs10407022) genes. These results suggest an association of an SNP in the *LIF* (rs929271) with low-prognosis groups, especially in patients younger than 35 years. Furthermore, an association of the A allele of *FSHR* (rs6166) with a suboptimal or poor ovarian response during conventional COH was observed, especially in patients with an adequate ovarian reserve.

### 3.3. Genotyping and Polymorphisms Analysis of the LIF Gene (rs929271) in Patients with Poor Response and Normal Responders

Patients in POSEIDON group 1 or 2 exhibited an unexpectedly poor or suboptimal ovarian response after standard ovarian stimulation, despite having adequate ovarian reserve parameters [15]. The results indicated that the *LIF* (rs929271) TG/GG genotypes and G allele were enriched in young patients and that the A allele frequency of *FSHR* (rs6166) was higher in POSEIDON groups 1 and 2. Thus, we compared the *LIF* (rs929271) and *FSHR* (rs6166) genotypes and allele frequencies of POSEIDON group 1 (*n* = 208) and group 2 (*n* = 361) with those of age-matched normal responders (age < 35 years, *n* = 391; age ≥ 35 years, *n* = 269, respectively).

The distribution of *LIF* (rs929271) and *FSHR* (rs6166) across the various ages and groups is presented in Table 3. The SNP in *LIF* (rs929271) results indicated significant differences between the genotypes of *LIF* (rs929271) in poor-responder groups and in young (age < 35 years) and normal responders; the TG/GG and G alleles was more common in group 1 (age < 35 years; TG/GG: 69.2%; G allele 42.3%) than in normal responders (TG/GG: 58.3%; G allele 36.3%; *p* = 0.0279 and 0.0425, respectively). This distribution did not significantly differ between group 2 (age ≥ 35 years) and normal responders (Table 3).

With regard to *FSHR* (rs6166), the A allele was more common in older women (group 2, 69.4%) than in normal responders (63.8%; *p* = 0.0354). However, the *FSHR* (rs6166) genotypes were not more common in women older than 35 years (Table 3). In women younger than 35 years, the distributions of *FSHR* (rs6166) allele frequency and genotypes were similar between patients with poor response and individuals in the control group.

Overall, these results demonstrated an association of SNPs in the *LIF* (rs929271) with poor response, especially in patients younger than 35 years. The allele frequencies of *FSHR* (rs6166) were associated with poor response, especially in women older than 35 years.

### 3.4. Association between Genotype and Ovarian Response

Our results revealed that the *LIF* (rs929271) TG/GG genotypes were more common in younger patients with poor response than in normal responders. The influence of *LIF* (rs929271) genotypes on clinical characteristics and clinical outcomes was investigated in patients younger than 35 years undergoing ART treatment.

A total of 599 women (age < 35 years) were included in genetic model analysis, and the results are displayed in Table 4. Of the 599 patients, 372 (62.1%) patients had TG/GG genotypes and 227 (37.9%) patients had TT genotypes.

The patients’ clinical characteristics between the TT and TG/GG genotypes of the *LIF* (rs929271) did not differ with respect to age; BMI; AMH; baseline FSH, LH, and E2; duration of infertility; E2 on human chorionic gonadotrophin (HCG) administration day; P4 on HCG administration day; number of D3 embryos; or D3 good embryo rate (Table 4). However, women with the *LIF* (rs929271) TG/GG genotype retrieved significantly fewer oocytes than those with the TT genotype (14 vs.16, *p* = 0.0109; Table 4). The *LIF* (rs929271) gene was also associated with a significantly lower number of mature oocytes for the genotype TG/GG than that for the TT genotype (11 vs. 13, *p* = 0.0082; Table 4). These results suggest that *LIF* (rs929271) may contribute to decreases in the number of oocytes retrieved and the number of mature oocytes in young women with infertility younger than 35 years undergoing ART treatment.

Because the allele frequencies of *FSHR* (rs6166) were associated with the POSEIDON group 2 patients (age ≥ 35 years), the effects of the *FSHR* (rs6166) genotypes on clinical characteristics and clinical outcomes were also examined in older patients (age ≥ 35 years) undergoing ART treatment.

A total of 630 women older than 35 years were included in genetic model analysis (Table 5); 564 (89.52%) of them had AA/AG genotypes and 66 (10.48%) had GG genotypes.

The patients’ clinical characteristics between the AA/AG and GG genotypes of the *FSHR* (rs6166) did not significantly differ with respect to age; BMI; AMH; baseline FSH, LH, and E2; duration of infertility; E2 on HCG administration day; P4 on HCG administration day; number of oocytes retrieved; number of D3 embryos; or D3 good embryo rate (Table 5). However, women with the *FSHR* (rs6166) AA/AG genotype had significantly fewer mature oocytes than women with a GG genotype (8 vs. 10, *p* = 0.0315; Table 5). This suggests that *FSHR* (rs6166) may lead to lower numbers of mature oocytes in older women with infertility (age ≥ 35 years) undergoing ART treatment.

## 4. Discussion

We first evaluated the distribution of four SNP polymorphisms in POSEIDON groups. We found that women with infertility under the age of 35 (POSEIDON group 1 and 3) were associated with a higher frequency of *LIF* (rs929271) TG/GG genotypes and G allele. A higher frequency of *FSHR* (rs6166) A allele was observed in the women with infertility with adequate ovarian reserve (POSEIDON group 1 and 2). Second, we compared the distribution of *LIF* (rs929271) and *FSHR* (rs6166) in patients with infertility with an adequate ovarian reserve (POSEIDON group 1 and 2) with normal responders. The women with infertility under the age of 35 (POSEIDON group 1) were associated with a higher frequency of TG/GG genotypes and G allele of *LIF* (rs929271). The older women with infertility (age ≥ 35 years; POSEIDON group 2) were associated with a higher A allele frequency of *FSHR* (rs6166). Finally, we demonstrated that *LIF* (rs929271) may lead to fewer oocytes retrieved and a lower number of mature oocytes in young women with infertility under the age of 35 years and the *FSHR* (rs6166) may contribute to fewer number of mature oocytes in older women with infertility (age ≥ 35) undergoing ART treatment. According to our results, *LIF* (rs929271) and *FSHR* (rs6166) were associated with a statistically significant reduction in ovarian response to controlled ovarian stimulation (COH) in younger and older women, respectively, with an adequate ovarian reserve. These results indicated that both *LIF* (rs929271) and *FSHR* (rs6166) might modulate ovarian response during COH.

One study demonstrated that GT/GG genotypes and the G allele of *LIF* (rs929271) are significantly enriched in patients with infertility under the age of 35 years, but not in older patients with unexplained infertility [25]. Our study also revealed a significantly higher rate of the TG/GG genotype and the G allele of *LIF* (rs929271) in younger patients with infertility (group 1 and 3) but not in older patients (group 2 and 4) among the patients with poor response.

Our previous study indicated that the frequencies of the SNP in *FSHR* (rs6166) were similar when we compared POSEIDON group 3 with group 4 (low ovarian reserve) [24]. However, when we analyzed the frequencies of the SNP in *FSHR* (rs6166) among POSEIDON groups, our current study indicated a higher frequency of A allele of *FSHR* (rs6166) in the women with infertility with adequate ovarian reserve (POSEIDON group 1 and 2; Table 2). In other words, among patients with poor response, women with infertility with adequate ovarian reserve exhibited a higher frequency of *FSHR* (rs6166) A allele than women with infertility with low ovarian reserve. However, for women with infertility with low ovarian reserve, the frequency of A allele of *FSHR* (rs6166) was not distributed differently.

The number of oocytes retrieved following COH for IVF/ICSI is closely related to cumulative live birth rates (LBR) after utilization of all fresh and frozen embryos [26]. High responders (>15 oocytes) and normal responders have a higher cumulative LBR (61.5%; 50.5%) than suboptimal (4–9 oocytes) and low responders (1–3 oocytes; 39.7%; 21.7%) [26]. POSEIDON groups 1 and 2 exhibited adequate ovarian reserves but had unexpectedly poor (<4 retrieved oocytes) or suboptimal responses (4–9 retrieved oocytes) to stimulation, leading to lower cumulative LBRs, compared with normal responders [14]. When we compared POSEIDON groups 1 and 2 with their age-matched normal responders, we found that the SNP in *LIF* (rs929271) was distributed differently in POSEIDON group 1 and the control group. A higher frequency of A allele of *FSHR* (rs6166) was noted in POSEIDON group 2 than the control group (Table 3). Our results indicated that these two SNPs may increase the likelihood of an unexpectedly lower ovarian response relative to the actual ovarian reserve.

With regard to the clinical characteristics and ovarian response of young patients (<35 years) with AMH ≥ 1.2 ng/mL, a genetic model analysis revealed that compared with a TT genotype, a TG or GG genotype in *LIF* (rs929271) was associated with a significantly lower number of oocytes (14 vs. 16) and mature oocytes (11 vs. 13). These results were reflective of the higher percentages of TG/GG genotypes (69.2%) in POSEIDON group 1 than in normal responders (58.3%) (Table 3). These effects could contribute to unexpected suboptimal or poor COH response in POSEIDON group 1.

With regard to the clinical characteristics and ovarian response of older patients (≥35 years) with AMH ≥ 1.2 ng/mL, a genetic model analysis revealed that compared with a GG genotype, an AA/AG genotype in *FSHR* (rs6166) was associated with a significantly lower number of mature oocytes (8 vs. 10). The higher frequency of A allele of POSEIDON group 2 than in normal responders (Table 3) may have contributed to the unexpectedly poor or suboptimal COH response in POSEIDON group 2.

Numerous studies have investigated the *LIF* polymorphisms among women younger than 35 years with unexplained infertility [25], as predictors of implantation efficiency and pregnancy outcomes [27] and in the prediction of recurrent implantation failure in combination with estrogen receptor 1 [28]. In animal models, LIF supports the primordial to primary follicle transition in rat ovaries [5], coordinates follicular growth in cultured murine ovarian tissues [29], enhances bovine oocyte maturation and early embryo development [6], and modulates gene and miRNA expression in bovine oocytes and embryos under in vitro maturation conditions [7]. LIF may modulate not only in implantation, but also in folliculogenesis. Women with infertility under 35 years who have TG or GG phenotypes of *LIF* (rs929271) may exhibit unexpectedly poor or suboptimal responses during COH, and LIF may affect folliculogenesis both in gonadotropin-independent growth and gonadotropin-dependent growth.

*FSHR* polymorphisms have been investigated in relation to ovarian response. The earliest report on this topic indicated that more FSH ampoules are required to reach successful stimulation when the G/G genotype of FSH (rs6166) is present at a significantly higher basal level [20]. Numerous studies have reported that patients with *FSHR* (rs6166) A/A and rs6165 G/G genotypes and rs1394205 A/A genotype tend to exhibit reduced ovarian response during COH and require higher FSH dosages [19,30,31,32,33,34]. One multicenter multinational prospective study examined the effect of polymorphisms in *FSHR* and FSHB genes on ovarian response and reported that the presence of *FSHR* SNPs (rs6165, rs6166, rs1394205) affected ovarian responses with a fixed dose of 150 IU rFSH [18]. These findings suggest that *FSHR* SNPs affect folliculogenesis in gonadotropin-dependent growth. Therefore, fewer oocytes are retrieved by older patients (≥35 years) with AA or AG genotypes of *FSHR* (rs6166) with *AMH* ≥ 1.2 ng/mL receiving ART treatment.

With regard to the clinical implications of our findings, *LIF* (rs929271) and *FSHR* (rs6166) analysis should be considered for young and older women with infertility who are expected to be normal responders but who exhibit an unexpectedly poor or suboptimal COH response.

It will be a challenge for reproductive specialists to predict impaired or poor ovarian response to exogenous gonadotropins when infertile women with an adequate ovarian reserve test (AFC ≥ 5–7 follicles or AMH ≥ 1.2 ng/mL) undergo their first ART. Our data showed that the women with infertility under the age of 35 (POSEIDON group 1) were associated with a higher frequency of TG/GG genotypes and G allele of *LIF* (rs929271). The older women with infertility (age ≥ 35 years; POSEIDON group 2) were associated with a higher A allele frequency of *FSHR* (rs6166). Infertile patients in POSEIDON group 1 and 2 have some common features, such as an adequate ovarian reserve test (AFC ≥ 5–7 follicles or AMH ≥ 1.2 ng/mL), but they revealed unexpected impaired or poor ovarian response to exogenous gonadotropins undergoing COH. Thereafter, we could consider genotyping *LIF* (rs929271) and *FSHR* (rs6166) among young and older women with infertility who are expected to be normal responders before they enter their first ART to avoid an unexpected or a hypo-response. This study has two major strengths: (a) the use of only one type of long GnRH agonist stimulation protocol and (b) the strict inclusion criteria for patients with poor response based on the POSEIDON criteria.

This study has several limitations. First, we included only Han Chinese people. Second, we did not include data regarding pregnancy outcomes after embryo transfer. Therefore, we cannot conclude that *LIF* (rs929271) or *FSHR* (rs6166) influences the rate of clinical pregnancy.

## 5. Conclusions

*LIF* (rs929271) and *FSHR* (rs6166) were associated with a statistically significant reduction in ovarian response to COH in younger and older women, respectively, with an adequate ovarian reserve, indicating that both *LIF* (rs929271) and *FSHR* (rs6166) might modulate ovarian response during COH. *LIF* (rs929271) and *FSHR* (rs6166) should be considered as potential biomarkers for poor or suboptimal COH responses among young and older women with infertility who are expected to be normal responders.

## Figures and Tables

**Table 1 jcm-12-00796-t001:** Genetic variation and primer sequence for studied SNPs.

Gene (SNP ID)	Variation	Region	Forward and Backward Primer Sequences
*GnRHR* (rs3756159)	G > A	Non-coding Intron	CCGACTTTCATAGCCACACCCTGAAT CACAACATGAAAGGTATAAAGCCCTCCAG
*FSHR* (rs6166)	2039 G > A Asn680Ser	Coding (exon)	CTTCAGCTCCCAGAGTCACC CATTGTGTTTTAGTTTTGGGCTAA
*AMH* (rs10407022)	146 T > G Ile49Ser	Coding (exon)	TCCGAGAAGACTTGGACTGG AGCTGCTGCCATTGCTGT
*LIF* (rs929271)	c.1414T > G	Non-coding Promoter	Reference to TagMan^®^ SNP genotyping system

*GnRHR*: gonadotropin-releasing hormone receptor, *FSHR*: follicle-stimulating hormone receptor, *AMH*: anti-Müllerian hormone, *LIF*: leukemia inhibitory factor, Asn: asparagine, Ser: serine, Ile: isoleucine.

**Table 2 jcm-12-00796-t002:** Distribution of SNP polymorphism in POSEIDON groups (*n* = 1084).

	POSEIDON Groups				*p* Value ^1^
	1 (*n* = 208)	2 (*n* = 361)	3 (*n* = 117)	4 (*n* = 398)	
*GnRHR* (rs3756159)					
GG	51(24.5%)	117(32.4%)	37(31.6%)	113(28.4%)	*p* = 0.3100
GA	118(56.7%)	178(49.3%)	55(47.0%)	196(49.2%)	
AA	39(18.8%)	66(18.3%)	25(21.4%)	89(22.4%)	
G	220(52.9%)	412(57.1%)	129(55.1%)	422(53.0%)	*p* = 0.3775
A	196(47.1%)	310(42.9%)	105(44.9%)	374(47.0%)	
*FSHR* (rs6166)					
AA	99(47.6%)	170(47.1%)	47(40.2%)	157(39.4%)	*p* = 0.1657
AG	92(44.2%)	161(44.6%)	59(50.4%)	191(48.0%)	
GG	17(8.2%)	30(8.3%)	11(9.4%)	50(12.6%)	
A	290(69.7%)	501(69.4%)	153(65.4%)	505(63.4%)	*p* = 0.0453 *
G	126(30.3%)	221(30.6%)	81(34.6%)	291(36.6%)	
*AMH* (rs10407022)					
TT	67(32.2%)	128(35.5%)	47(40.2%)	149(37.4%)	*p* = 0.8423
TG	100(48.1%)	167(46.3%)	51(43.6%)	176(44.2%)	
GG	41(19.7%)	66(18.3%)	19(16.2%)	73(18.3%)	
T	234(56.3%)	423(58.6%)	145(62.0%)	474(59.5%)	*p* = 0.5164
G	182(43.7%)	299(41.4%)	89(38.0%)	322(40.5%)	
*LIF* (rs929271)					
TT	64(30.8%)	154(42.7%)	42(35.9%)	163(41.0%)	*p* = 0.0100 *
TG	112(53.8%)	158(43.8%)	50(42.7%)	192(48.2%)	
GG	32(15.4%)	49(13.6%)	25(21.4%)	43(10.8%)	
T	240(57.7%)	466(64.5%)	134(57.3%)	518(65.1%)	*p* = 0.0156 *
G	176(42.3%)	256(35.5%)	100(42.7%)	278(34.9%)	

*GnRHR*: gonadotropin-releasing hormone receptor, *FSHR*: follicle stimulating hormone receptor, *AMH*: anti-Müllerian hormone, *LIF*: leukemia inhibitory factor. ^1^ by Chi-squared test, * Significance was indicated if *p* < 0.05.

**Table 3 jcm-12-00796-t003:** Distribution of *LIF* (rs929271) and *FSHR* (rs6166) in older (≥ 35 years, *n* = 630) and younger (<35 years, *n* = 599) patients with AMH ≥ 1.2 ng/mL.

	Groups of Response		*p* Value ^1^
**≥35 Y/O**	**POSEIDON 2 (*n* = 361)**	**Normal Response (*n* = 269)**	
*LIF* (rs929271)			
TT	154(42.7%)	102(37.9%)	*p* = 0.4781
TG	158(43.8%)	126(46.8%)	
GG	49(13.6%)	41(15.2%)	*p* = 0.2436
T	466(64.5%)	330(61.3%)	
G	256(35.5%)	208(38.7%)	
*FSHR* (rs6166)			
AA	170 (47.1%)	110 (40.9%)	*p* = 0.0757
AG	161 (44.6%)	123 (45.7%)	
GG	30 (8.3%)	36 (13.4%)	
A	501(69.4%)	343 (63.8%)	*p* = 0.0354 *
G	221(30.6%)	195 (36.2%)	
**<35 Y/O**	**POSEIDON 1 (*n* = 208)**	**Normal Response (*n* = 391)**	
*LIF* (rs929271)			
TT	64(30.8%)	163(41.7%)	*p* = 0.0279 *
TG	112(53.8%)	172(44.0%)	
GG	32(15.4%)	56(14.3%)	
T	240 (57.7%)	498(63.7%)	*p* = 0.0425 *
G	176 (42.3%)	284(36.3%)	
*FSHR* (rs6166)			
AA	99 (47.6%)	171 (43.7%)	*p* = 0.3834
AG	92 (44.2%)	175 (44.8%)	
GG	17 (8.2%)	45 (11.5%)	
A	290 (69.7%)	517 (66.1%)	*p* = 0.2061
G	126 (30.3%)	265 (33.9%)	

*LIF*: leukemia inhibitory factor, *FSHR*: follicle-stimulating hormone receptor. ^1^ by Chi-squared test, * Significance was indicated if *p* < 0.05.

**Table 4 jcm-12-00796-t004:** Clinical characteristics for younger patients (<35 years) with AMH ≥ 1.2 ng/mL (*n* = 599) undergoing ART treatment according to genotype of *LIF* SNP rs929271.

*LIF* rs929271	TT (*n* = 227)		TG/GG (*n* = 372)		
	Median	25%–75%	Median	25%–75%	*p* ^1^
Age (years)	32.0	29.0 to 33.0	31.0	30.0 to 33.0	0.8501
BMI (kg/m^2^)	21.5	19.7 to 23.8	21.1	19.55 to 23.67	0.3258
AMH (ng/mL)	4.90	2.94 to 8.35	4.68	2.87 to 8.11	0.6640
Baseline FSH (IU/L)	6.40	4.56 to 7.77	6.11	4.57 to 7.80	0.5976
Baseline LH (IU/L)	5.03	3.50 to 8.50	5.3	3.24 to 8.10	0.4799
Baseline E2 (ng/mL)	28.0	19.0 to 48.0	27.0	19.0 to 49.5	0.8584
Duration of Infertility (years)	2.0	1.2 to 4.0	2.5	1.43 to 4.0	0.5786
E2 on HCG day (ng/mL)	2686.0	1752.5 to 4098.5	2784.0	1795.8 to 4428.8	0.4896
P4 on HCG day (pg/mL)	1.14	0.79 to 1.51	1.14	0.74 to 1.62	0.7889
Oocytes number	16	11 to 22	14	9 to 20	0.0109 *
MII number	13	9 to 18	11	7 to 16	0.0082 **
Number of Day3 Embryos	11	7 to 15	10	6 to 15	0.0904
Day3 Good Embryo Rate (%)	53.9	37.5 to 69.2	55.0	37.500 to 70.000	0.9984

*LIF*: leukemia inhibitory factor, BMI: body mass index, AMH: anti-Müllerian hormone, FSH: follicle-stimulating hormone, LH: luteinizing hormone, E2: estradiol, HCG: human chorionic gonadotropin, P4: progesterone, MII: metaphase II oocyte. ^1^ by Mann–Whitney U test, * Significance was indicated if *p* < 0.05, ** Significance was indicated if *p* < 0.01.

**Table 5 jcm-12-00796-t005:** Clinical characteristics for older patients (≥35 years) with AMH ≥ 1.2 ng/mL (*n* = 630) undergoing ART treatment according to genotype of *FSHR* SNP (rs6166).

*FSHR* (rs6166)	AA/AG (*n* = 564)		GG (*n* = 66)		
	Median	25%–75%	Median	25%75%	*p* ^1^
Age (years)	38.0	36.0 to 39.0	37.0	36.0 to 39.0	0.4614
BMI (kg/m^2^)	21.7	20.0 to 24.2	22.3	19.7 to 25.1	0.9450
AMH (ng/mL)	3.07	1.94 to 5.31	3.29	2.13 to 5.44	0.3255
Baseline FSH (IU/L)	6.30	4.40 to 8.10	6.56	4.52 to 8.40	0.2727
Baseline LH (IU/L)	4.70	3.08 to 6.82	4.70	3.60 to 6.20	0.6209
Baseline E2 (ng/mL)	28.0	19.0 to 53.0	25.0	18.0 to 67.0	0.5810
Duration of Infertility (years)	3.0	2.0 to 5.0	3.5	2.0 to 6.0	0.6520
E2 on HCG day (ng/mL)	1993.0	1144.5 to 3146.3	2241.0	1447.0 to 3198.0	0.3316
P4 on HCG day (pg/mL)	0.99	0.63 to 1.40	1.07	0.66 to 1.40	0.7464
Oocytes number	11	6 to 16	13	8 to 16	0.1439
MII number	8	5 to 13	10	6 to 14	0.0315 *
Number of Day3 Embryos	7	4 to 12	10	5 to 14	0.0670
Day3 Good Embryo Rate (%)	55.6	38.890 to 71.430	56.3	40.0 to 66.7	0.6739

*FSHR*: follicle-stimulating hormone receptor, BMI: body mass index, AMH: anti-Müllerian hormone, FSH: follicle stimulating hormone, LH: luteinizing hormone, E2: estradiol, HCG: human chorionic gonadotropin, P4: progesterone, MII: metaphase II oocyte. ^1^ by Mann–Whitney U test, * Significant was indicated if *p* < 0.05.

## Data Availability

The data presented in this study are available on request from the corresponding author.

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
