# Peer review of "Association between Leukemia Inhibitory Factor Gene Polymorphism and Clinical Outcomes among Young Women with Poor Ovarian Response to Assisted Reproductive Technology"

_jcm, 2023, doi:10.3390/jcm12030796_

Round 1

Reviewer 1 Report

Dear respected editor;

Regarding the article number JCM-2112522, titled; Association Between Leukemia Inhibitory Factor Gene Polymorphism and Clinical Outcomes Among Young Women with Poor Ovarian Response to Assisted Reproductive Technology

This study aimed to investigate the influence of four SNPs (LIF, AMH, FSHR, and GnRHR) on stimulation phase ovarian response and clinical outcomes among patients with infertility undergoing their first ovarian stimulation for in vitro fertilization or intracytoplasmic sperm injection.

General points to be considered

-          Title: good  

-         Abstract: well written

- This research is well-designed and written.

1.     Introduction:

Lines 50-54, spaces before the sign should be minimized

However, some patients whose adequate 49 ovarian reserve parameters are at acceptable levels (AFC ≥ 5; AMH ≥ 1.2 ng/mL) exhibit 50 an unexpectedly poor or suboptimal ovarian response after standard ovarian stimulation, 51 resulting in a lower cumulative delivery rate than that of normal responders [7]. These 52 patients were categorized into group 1 (young patients <35 years) or group 2 (older pa- 53 tients ≥ 35 years) on the basis of the POSEIDON criteria [8].

The fourth paragraph in the introduction should be moved to be the first paragraph.

Leukemia inhibitory factor (LIF) is a cytokine belonging to the interleukin-6 super-family. It was first identified for its ability to induce macrophage differentiation of murine myeloid leukemia cells and inhibit their proliferation [15,16]. LIF modulates various functions and plays important roles in embryo implantation and in nonhormonal contraception [17]. LIF expression has been detected in human follicular fluid and ovarian stromal cells [18]. In animal models, LIF has been reported to enhance the primordial to primary follicle transition of in vitro cultured rat ovaries [19] and promote bovine oocyte maturation for the in vitro maturation of denuded bovine oocytes [20,21], indicating LIF’s potential involvement in folliculogenesis. We hypothesize that LIF is involved in ovarian sensitivity to gonadotropin stimulation, particularly in individuals exhibiting an unexpectedly poor or suboptimal response.

2.     Materials and Methods

2.1. Study design, and setting

This is a case-control study including couples with infertility undergoing their first 83 ovarian stimulation for IVF or ICSI at any period from January 2014 to December 2015.

- I think this study is considered a case-control cross sectional study

- Its preferred to add the exclusion criteria.

3.     Results

In all tables, all abbreviations must be defined in their full names and must be written under the table.

Space should be there before line 147 under table 1.

Statistical values presented in tables should be defined under every table and the type of used test should be mentioned.

4.     Discussion

-          Well written.

-          Still need addition of some information about the clinical and diagnostic importance of the studied genes.

5.     Conclusion

It is very short and need more clarification (slightly improvement).

References:

Well written according to the journal instructions.

Need revision and should be optimized.

For example; reference 6, page numbers should be added. 8, 21, 24, and 29 need to be checked

Author Response

Response to Reviewer 1 Comments

Regarding the article number JCM-2112522, titled; Association Between Leukemia Inhibitory Factor Gene Polymorphism and Clinical Outcomes Among Young Women with Poor Ovarian Response to Assisted Reproductive Technology

This study aimed to investigate the influence of four SNPs (LIF, AMH, FSHR, and GnRHR) on stimulation phase ovarian response and clinical outcomes among patients with infertility undergoing their first ovarian stimulation for in vitro fertilization or intracytoplasmic sperm injection.

Point 1: Introduction:

  • Lines 50-54, spaces before the sign should be minimized
  • However, some patients whose adequate 49 ovarian reserve parameters are at acceptable levels (AFC ≥ 5; AMH ≥ 1.2 ng/mL) exhibit 50 an unexpectedly poor or suboptimal ovarian response after standard ovarian stimulation, 51 resulting in a lower cumulative delivery rate than that of normal responders [7]. These 52 patients were categorized into group 1 (young patients ï¼œ35 years) or group 2 (older pa- 53 tients ≥ 35 years) on the basis of the POSEIDON criteria [8].

Response 1: Thank you for the suggestion. We had made correction in the Introduction section of the revised manuscript.

Point 2: The fourth paragraph in the introduction should be moved to be the first paragraph.

  • Leukemia inhibitory factor (LIF) is a cytokine belonging to the interleukin-6 super-family. It was first identified for its ability to induce macrophage differentiation of murine myeloid leukemia cells and inhibit their proliferation [15,16]. LIF modulates various functions and plays important roles in embryo implantation and in nonhormonal contraception [17]. LIF expression has been detected in human follicular fluid and ovarian stromal cells [18]. In animal models, LIF has been reported to enhance the primordial to primary follicle transition of in vitro cultured rat ovaries [19] and promote bovine oocyte maturation for the in vitro maturation of denuded bovine oocytes [20,21], indicating LIF’s potential involvement in folliculogenesis. We hypothesize that LIF is involved in ovarian sensitivity to gonadotropin stimulation, particularly in individuals exhibiting an unexpectedly poor or suboptimal response.

Response 2: Thank you for the suggestion. We had moved the fourth paragraph to the first paragraph in the Introduction section of the revised manuscript.

Point 3: Materials and Methods

  • 1. Study design, and setting
    • This is a case-control study including couples with infertility undergoing their first 83 ovarian stimulation for IVF or ICSI at any period from January 2014 to December 2015.
    • - I think this study is considered a case-control cross sectional study

Response 3: Thank you for recommendation. We made the revision “This is a case-control crsoo sectional study” in the Materials and Methods section of the revised manuscript.

Point 4: - Its preferred to add the exclusion criteria.

Response 4: Thank you for recommendation. We added the exclusion criteria in the revision in the Materials and Methods section of the revised manuscript. “Exclusion criteria were genetic anomalies, autoimmune dysfunction, inflammatory disease and other systemic disorders.”

Point 5: - Results

  • In all tables, all abbreviations must be defined in their full names and must be written under the table.

Response 5: Thank you for recommendation. In all table, abbreviations have been defined in their full names and written under the table in the revised manuscript.

Point 6: - Space should be there before line 147 under table 1.

Response 6: Thank you for recommendation. Space has been added.

Point 7: - Statistical values presented in tables should be defined under every table and the type of used test should be mentioned.

Response 7: Thank you for suggestion. Statistical values presented in tables have been defined under every table and the type of used test has been mentioned in the revised manuscript.

Point 8: - Discussion

  • Well written
  • Still need addition of some information about the clinical and diagnostic importance of the studied genes.

Response 8: Thank you for recommendation. We added some information about the clinical and diagnostic importance of our studied genes in the Discussion section of the revised manuscript, as follow: “It will be a challenge for reproductive specialists to predict impaired or poor ovarian response to exogenous gonadotropins when infertile women with an adequate ovarian reserve test (AFC≥ 5–7 follicles or AMH≥ 1.2 ng/mL) undergo their first ART. Our data showed that the women with infertility under the age of 35 (POSEIDON group 1) were associated with a higher frequency of TG/GG genotypes and G allele of LIF (rs929271). The older women with infertility (age ≥ 35 years; POSEIDON group 2) were associated with a higher A allele frequency of FSHR (rs6166). Infertile patients in POSEIDON group 1 and 2 have some common features, such as with an adequate ovarian reserve test (AFC≥ 5–7 follicles or AMH≥ 1.2 ng/mL) but revealed unexpected impaired or poor ovarian response to exogenous gonadotropins undergoing COH. Thereafter, we could consider to genotype LIF (rs929271) and FSHR (rs6166) among young and older women with infertility who are expected to be normal responders before they enter their first ART to avoid an unexpected or a hypo-response.”

Point 9: Conclusion

  • It is very short and need more clarification (slightly improvement).

Response 9: Thank you for recommendation. We modified our conclusion as follow: “LIF (rs929271) and FSHR (rs6166) were associated with a statistically significant reduction in ovarian response to COH in younger and older women, respectively, with an adequate ovarian reserve, indicating that both LIF (rs929271) and FSHR (rs6166) might modulate ovarian response during COH. LIF (rs929271) and FSHR (rs6166) should be considered as potential biomarkers for poor or suboptimal COH responses among young and older women with infertility who are expected to be normal responders.”

Point 10: References:

  • Well written according to the journal instructions.
  • Need revision and should be optimized.
  • For example; reference 6, page numbers should be added. 8, 21, 24, and 29 need to be checked

Response 10: Thank you for recommendation. After we moved the fourth paragraph to the first paragraph in the Introduction section of the revised manuscript. Reference 6 became Reference 13; Reference 8 became Reference 15; Reference 21 became Reference 7; Reference 24 was still Reference 24 and Reference 29 was still Reference 29. We added page numbers in these five References based on their journal format.

Thank you. Your recommendation is very helpful for our study and manuscript. Your kindness in helping us to publish this manuscript will be much appreciated.

All authors have approved the manuscript and agree with its submission to Journal of Clinical Medicine.

Sincerely yours,

Tsung-Hsien Lee, M.D. Ph.D.

Reviewer 2 Report

Comments 

The abstract should list the main results, for example, the results observed with follicle stimulating hormone receptor (FSHR) are missing. The first paragraph of the discussion synthesizes the main results very well, these ideas could be taken for the abstract.

Correct editorial details, e.g.

Remove unnecessary adverbs, e.g., line 188 "these results clearly demonstrated an association of SNPs in LIF (rs929271) with poor response" instead use "these results demonstrated an association of SNPs in LIF (rs929271) with poor response."

Line 189. "However, FSHR (rs6166) allele frequencies were" The adversative nexus However is not well used as it is not indicating the opposition of two ideas. 

Author Response

Response to Reviewer 2 Comments

Point 1: The abstract should list the main results, for example, the results observed with follicle stimulating hormone receptor (FSHR) are missing. The first paragraph of the discussion synthesizes the main results very well, these ideas could be taken for the abstract.

Response 1: Thank you for recommendation. We revised our abstract to include the results of FSHR.

  • Abstract: Background: Does the presence of single nucleotide polymorphisms (SNPs) in the leukemia inhibitory factor (LIF) gene affect ovarian response in infertile young women? Methods: This was a case-control study recruiting 1744 infertile women between January 2014 to December 2015. The 1084 eligible patients were stratified into 4 groups using the POSEIDON criteria. The gonadotropin-releasing hormone receptor (GnRHR), follicle stimulating hormone receptor (FSHR), anti-Müllerian hormone (AMH), and LIF SNP genotypes were compared among the groups. The distributions of LIF and FSHR among younger and older patients were compared. Clinical outcomes were also compared. Results: The 4 groups of poor responders had different distributions of SNP in LIF. The prevalence of LIF genotypes among young poor ovarian responders differed from those of normal responders. Genetic model analyses in infertile young women revealed that the TG or GG genotype in the LIF resulted in fewer oocytes retrieved and fewer mature oocytes relative to the TT genotypes. In older women, the FSHR SNP genotype contributed to fewer numbers of mature oocytes. Conclusions: LIF and FSHR SNP genotypes were associated with a statistically significant reduction in ovarian response to controlled ovarian hyperstimulation in younger and older women with an adequate ovarian reserve, respectively.

Point 2: Correct editorial details, e.g.

  • Remove unnecessary adverbs, e.g., line 188 "these results clearly demonstrated an association of SNPs in LIF (rs929271) with poor response" instead use "these results demonstrated an association of SNPs in LIF (rs929271) with poor response."

Response 2: Thank you for recommendation. We had removed this unnecessary adverb, “clearly”.

Point 3: Line 189. "However, FSHR (rs6166) allele frequencies were" The adversative nexus However is not well used as it is not indicating the opposition of two ideas.

Response 3: Thank you for recommendation. We had removed “However” in the revision in the Result section of the revised manuscript.

Thank you. Your recommendation is very helpful for our study and manuscript. Your kindness in helping us to publish this manuscript will be much appreciated.

All authors have approved the manuscript and agree with its submission to Journal of Clinical Medicine.

Sincerely yours,

Tsung-Hsien Lee, M.D. Ph.D.